# Composite ADRC Speed Control Method Based on LTDRO Feedforward Compensation

**DOI:** 10.3390/s24082605

**Published:** 2024-04-19

**Authors:** Rencheng Jin, Junwei Wang, Yangyi Ou, Jianzhang Li

**Affiliations:** Key Laboratory for Micro/Nano Technology and System of Liaoning Province, Dalian University of Technology, Dalian 116024, China; dluter2021@mail.dlut.edu.cn (J.W.); oyy@mail.dlut.edu.cn (Y.O.); lijianzhang@mail.dlut.edu.cn (J.L.)

**Keywords:** stepper motor, feedforward control, active disturbance rejection control, dimensionality reduction observer

## Abstract

The performance of the extended state observer (ESO) in an Active Disturbance Rejection Control (ADRC) is limited by the operational load in stepper motor control, which has high real-time requirements and may cause delays. Additionally, the complexity of parameter tuning, especially in high-order systems, further limits the ESO’s performance. This paper proposes a composite ADRC (LTDRO-ADRC) based on a load torque dimensionality reduction observer (LTDRO). Firstly, the LTDRO is designed to estimate abrupt load disturbances that are difficult to compensate for using the ESO. Secondly, the transfer function under the double-closed loop is deduced. Additionally, the LTDRO uses a magnetic encoder to gather the system state and calculate the load torque. It then outputs a compensating current feedforward to the current loop input. This method reduces the delay and complexity of the ESO, improving the response speed of the ADRC speed ring and the overall response of the system to load changes. Simulation and experimental results demonstrate that it significantly enhances dynamic control performance and steady-state errors. LTDRO-ADRC can stabilize the speed again within 49 ms and 17 ms, respectively, in the face of sudden load increase and sudden load removal. At the same time, in terms of steady-state error, compared with ADRC and CADRC, they have increased by 94% and 88%, respectively. In terms of zero-speed starting motors, the response speed is increased by 58% compared to a traditional ADRC.

## 1. Introduction

Stepper motors, due to their straightforward architecture, high open-loop control accuracy, low cost of drivers, and ease of speed regulation, are extensively utilized in various industrial applications, including robotic arms, CNC machine tools, and new energy electric vehicles. Nonetheless, the application in robotic arms exposes stepper motors to challenges such as modeling inaccuracies, load disturbances, and other factors, leading to potential step losses, diminished anti-interference capability, and torque variability. These issues compromise not only the steady-speed precision of stepper motors but also the overall performance of the robotic arm control system. Given that traditional PID and ADRC strategies fall short of addressing the performance demands of current robotic arm joint stepper motors, the quest for advanced steady-speed motor control algorithms has emerged as a prominent research focus in recent years [1]. To counter the stepper motors’ reduced anti-interference efficiency under abrupt load shifts, the prevailing approach is the implementation of closed-loop control to bolster performance. This approach bifurcates into linear and nonlinear control methods. Linear control modalities typically integrate traditional PID control with Field-Oriented Control (FOC) technology, tailoring the current phase and magnitude for the two-phase hybrid stepper motor stator windings [2]. Research [3] has explored the efficacy of stepper motor FOC control through position step experiments employing a PI and ADRC in the current, speed, and position loops, highlighting the ADRC’s enhanced load damping capabilities. Further, study [4] has adapted ADRC for the motor’s current loop and attempts to optimize the ESO. Additionally, work [5] has introduced a dual closed-loop architecture featuring a PI speed loop and an ADRC position loop, proposing a method for ESO performance evaluation and a bias self-coupling compensation strategy to augment system robustness.

A traditional Active Disturbance Rejection Control (ADRC) is characterized by its reliance on numerous parameters, notably within the Extended State Observer (ESO), particularly in higher-order systems where system performance is critically dependent on the precise configuration and settings of the ESO’s structure and parameters. To mitigate these challenges, scholars have leveraged existing algorithms for the fine-tuning of ADRC parameters. A self-tuning decoupled controller design, utilizing the Genetic Algorithm (GA) for the optimization of ADRC parameters, was proposed by reference [6], significantly improving the system’s decoupling control performance. Reference [7] employed the TD3 algorithm for the dynamic adjustment of controller parameters, thereby achieving innovative trajectory tracking for aircraft. GAADRC, introduced by reference [8], harnesses the genetic algorithm for the tuning of motor ADRC parameters through a multi-objective optimization-based target function, culminating in the identification of optimal ADRC control parameters after several iterations. An innovative ADRC design, grounded in IMA and aiming for the determination of optimal ADRC parameters, was unveiled by reference [9]. References [10,11] based on Particle Swarm Optimization (PSO) proposed IPSO and APSO respectively to simplify the parameters of ADRC.

The ESO is tasked with real-time estimation of both system states and unknown disturbances, which imposes significant computational overhead and presents challenges for motor control systems requiring high real-time performance. As research progresses, efforts have been made to optimize the structure of the ESO while integrating it with other observers or algorithmic models for compensation, leading to the development of Composite ADRC. This approach aims to maintain system performance while minimizing the computational burden of the ESO as much as possible.

References [12,13] analyze the limitations of a conventional Linear ADRC and proposed methods to improve ADRCs based on Model Predictive Compensation. Reference [14] removes the tracking differentiator (TD) in the first-order ADRC and reference [15] optimizes the Luenberger Disturbance Observer to develop an ADRC-LOC controller suitable for linear uncertain disturbance systems of any order. Reference [16] introduced the IPSO-BP algorithm. It utilizes IDA-PBC to establish a PCHD model of the motor, incorporating the BP algorithm to update the parameters of the ADRC in real time. Reference [17] designed a novel current loop and proposed a Sliding Mode ADRC to regulate motor speed, which not only retains the original characteristics of ADRC but also ensures a smooth transition of ADRC parameters. Reference [18] combined the advantages of LADRC and NLADRC, proposing a Switching Control (SADRC) strategy. Reference [19] utilized a nonlinear Phase-Locked Loop (PLL) combined with special nonlinear functions for estimating uncertainties and load disturbances in actual conditions, designing a novel NPLLO structure and proposing an ADRC controller based on the new Nonlinear Phase-Locked Loop Observer (NPLLO). Reference [20] used interpolation fitting to reconstruct ESO and NLSEF to optimize ADRC. Reference [21] designed a reduced-order ESO and proposed a composite ADRC containing acceleration feedforward. Reference [22] improved the ESO to handle non-decreasing second-order differentiable disturbances to achieve minimal estimation error of the system and used the Lyapunov method to prove it. Reference [23] introduced an improved ESO, which is beneficial for achieving high-performance current control. Reference [24] presented a hybrid algorithm, combining a Frequency-Locked Loop (FLL) with an enhanced Second-Order Generalized Integrator (SOGI) and a Phase-Locked Loop (PLL) based on an ADRC to reduce rotor position estimation errors caused by higher-order harmonics. Reference [25] proposed Fuzzy-ADRC, enhancing its disturbance compensation capability. Reference [26] studied a composite ADRC for the second-order speed loop and position loop, proposing an IADRC to enhance system tracking accuracy. In reference [27], a bicyclic ADRC with position-velocity parameters that can be self-adaption is proposed for mechanical arm joints and applied to SCARA. Reference [28] enhances the system’s vibration suppression capabilities by designing a reduced-order extended state observer (RESO), while reference [29] addresses issues such as uncertain disturbances in flexible-joint manipulators (FJMs). Based on the RESO, a novel composite control method is devised, significantly improving the tracking performance.

Subsequent studies, such as those by references [16] through [27], have introduced various algorithmic enhancements and novel approaches, including the integration of genetic algorithms, particle swarm optimization, sliding mode control, and fuzzy logic, to address specific challenges in ADRC implementation, thereby advancing the robustness, adaptability, and performance of ADRC systems.

Existing studies commonly use observer techniques such as Luenberger observer, sliding mode observer, and Kalman filter. Although these studies make significant theoretical contributions, their practical applications can be complex and computationally heavy. The Load Torque Observer (LTO) design method based on the dimensionality reduction observer principle is simple to implement and has low computational complexity. It can be effectively combined with a high-order ESO and the two techniques complement each other’s advantages, reducing the system computational complexity and ESO delay. As shown in Table 1.

Specifically, a traditional ADRC has many parameters, and the ESO, as the core of ADRC control, will undoubtedly increase the burden on estimation and measurement tasks while narrowing the scope of uncertainty for unknown controlled objects; if the estimation capability does not meet the system requirements, requiring higher accuracy will actually extend the system’s estimation time, thereby severely affecting the performance of the active disturbance rejection control system. Especially for high-order systems, due to the difficulty of parameter adjustment and optimization, the ability to identify parameters may be reduced. At the same time, the total disturbance of the system cannot be effectively compensated, which limits the excellent control performance of the active disturbance rejection system. Therefore, the method of compensating part of the system model can, on the one hand, obtain more information about the controlled object, and on the other hand, it also reduces the burden on the controller to estimate the object and perform compensation. The load torque observer is designed using the dimensionality reduction observer principle to share the system burden. Through theoretical analysis, it can be proved that the dimensionality reduction observer has a small amount of calculations, compensates for the computational delay of high-order ESO well, and greatly improves the overall efficiency of the system.

This paper proposes a composite ADRC method for stepper motors combined with the Load Torque Dimensionality Reduction Observer (LTDRO). First, the closed-loop vector control and the mathematical model of the motor are introduced, and the speed loop is controlled by the ADRC. Secondly, in Section 3, the use of magnetic encoders to obtain system state variables such as motor position and speed is introduced, and the LTDRO is constructed. The observer can estimate external load disturbances in real time and output a feedforward compensation current after internal calculation. The ESO is used to compensate for all disturbances except load torque. In Section 4, the simulation and experimental results are presented. Finally, in Section 5, the effectiveness of the method is summarized.

## 2. Vector Control Mathematical Derivation

### 2.1. Derivation of Rest Coordinate Formula

The mathematical model in a stationary coordinate system primarily consists of three parts:(1)Voltage Equation:

Combining the winding inductance formula yields:(1)$$\left\{\begin{array}{c} U_{A}=r_{A}i_{A}+\left(L_{0}-L_{2}\cos2\theta_{e}\right)\frac{di_{A}}{dt}-L_{2}\sin2\theta_{e}\frac{di_{B}}{dt}\\ \;+2\omega_{L}L_{2}\left(i_{A}sin2\theta_{e}-i_{B}cos2\theta_{e}\right)-k_{e}\omega_{r}sin\theta_{e}\\ U_{B}=r_{B}i_{B}+\left(L_{0}+L_{2}\cos2\theta_{e}\right)\frac{di_{B}}{dt}-L_{2}\sin2\theta_{e}\frac{di_{A}}{dt}\\ \;-2\omega_{L}L_{2}\left(i_{B}sin2\theta_{e}+i_{A}cos2\theta_{e}\right)+k_{e}\omega_{r}sin\theta_{e} \end{array}\right.$$

In the formula, $U_{A},\;r_{A},\;U_{B},\;r_{B}$ represent the phase voltage and resistance of the stator winding of the motor, respectively; $\omega_{r}$ represents the mechanical angular speed of the rotor; $k_{e}$ is the back electromotive force coefficient.

(2)Mechanical Motion Equation:


(2)
$$T_{e}=J\frac{d\omega_{r}}{dt}+B\omega_{r}+T_{L}$$


In the formula, $T_{e}$ signifies the electromagnetic torque, J denotes the inertia moment, and $T_{L}$ is indicative of the torque exerted by the load.

(3)Torque Equation:

The total electromagnetic torque for a two-phase hybrid stepping motor is composed of the main electromagnetic torque $T_{m}$ and the detent torque $T_{s}$. By combining both, the electromagnetic torque can be expressed as:(3)$$T_{e}=T_{m}+T_{s}=N_{r}M_{s}rI_{m}\left(-i_{A}\sin\theta_{e}+i_{B}\cos\theta_{e}\right)+N_{r}L_{2}\left(i_{A}^{2}-i_{B}^{2}\right)\sin2\theta_{e}-2i_{A}i_{B}\cos2\theta_{e}$$

In summary, the equations for a two-phase hybrid stepper motor in a stationary coordinate system are as follows:(4)$$\left\{\begin{array}{l} U_{A}=r_{A}i_{A}+\left(L_{0}-L_{2}\cos2\theta_{e}\right)\frac{di_{A}}{dt}-L_{2}\sin2\theta_{e}\frac{di_{B}}{dt}\\ \;+2\omega_{L}L_{2}\left(i_{A}\sin2\theta_{e}-i_{B}\cos2\theta_{e}\right)-k_{e}\omega_{r}\sin\theta_{e}\\ U_{B}=r_{B}i_{B}+\left(L_{0}+L_{2}\cos2\theta_{e}\right)\frac{di_{B}}{dt}-L_{2}\sin2\theta_{e}\frac{di_{A}}{dt}\\ \;-2\omega_{L}L_{2}\left(i_{B}\sin2\theta_{e}+i_{A}\cos2\theta_{e}\right)+k_{e}\omega_{r}\sin\theta_{e}\\ T_{e}=J\frac{d\omega_{r}}{dt}+B\omega_{r}+T_{L}\\ T_{e}=T_{m}+T_{s}=N_{r}M_{s}rI_{m}\left(-i_{A}\sin\theta_{e}+i_{B}\cos\theta_{e}\right)\\ \;+N_{r}L_{2}\left(\left(i_{A}^{2}-i_{B}^{2}\right)\sin2\theta_{e}-2i_{A}i_{B}\cos2\theta_{e}\right) \end{array}\right.$$

### 2.2. Derivation of the Formula for d-q Coordinates

The conversion relationship between the rotating d-q coordinate system and the stationary α−β coordinate system is depicted in Figure 1.

Deduced from the previous analysis, this shows the angle separating the α-axis from the d-axis is equivalent to the electrical angle.

Conversion of stator currents $i_{\alpha },\;i_{\beta }$ from the stationary to the rotating coordinate system is executed via the Park transformation:(5)$$\left[\begin{array}{c} i_{\alpha }\\ i_{\beta } \end{array}\right]=\left[\begin{array}{cc} cos\theta_{e} & -sin\theta_{e}\\ sin\theta_{e} & cos\theta_{e} \end{array}\right]\left[\begin{array}{c} i_{d}\\ i_{q} \end{array}\right]$$

The process of reverting $i_{d},\;i_{q}$ from the rotating to the stationary coordinate system is conducted through the reverse Park transformation:(6)$$\left[\begin{array}{c} i_{d}\\ i_{q} \end{array}\right]=\left[\begin{array}{cc} cos\theta_{e} & sin\theta_{e}\\ -sin\theta_{e} & cos\theta_{e} \end{array}\right]\left[\begin{array}{c} i_{\alpha }\\ i_{\beta } \end{array}\right]$$

Integrating the equations for the two-phase magnetic flux of the motor from both stationary and rotating coordinate systems allows for the derivation of the voltage equation in the d−q rotating coordinate system as:(7)$$\left\{\begin{array}{c} U_{d}=L_{d}\frac{di_{d}}{dt}-L_{q}i_{q}\omega_{e}+i_{d}R\\ {\;U}_{q}=L_{q}\frac{di_{q}}{dt}+L_{d}i_{d}\omega_{e}+i_{q}R+M_{sr}I_{m}\omega_{e} \end{array}\right.$$

The formula for electromagnetic torque within the d−q rotating coordinate for the motor is:(8)$$T_{e}=N_{r}\cdot 2\left(L_{d}-L_{q}\right)i_{d}i_{q}+N_{r}I_{m}M_{sr}i_{q}$$

In vector control, there are three commonly used methods. This paper adopts the $i_{d}=0$ control, which simplifies the electromagnetic torque equation to:(9)$$T_{e}=N_{r}I_{m}M_{sr}i_{q}$$

In summary, when $i_{d}=0$, the electromagnetic torque is only dependent on the magnitude and direction of $i_{q}$. By controlling the magnitude and direction of $i_{q}$ appropriately, it is possible to control the electromagnetic torque of the two-phase hybrid stepper motor. This method, as utilized in this paper, achieves simplicity and avoids the demagnetizing effect. However, this control method also has its drawbacks. Setting $i_{d}$ to zero means that the reactive torque of the motor cannot be utilized. This results in low utilization of stator current, requiring larger currents to generate the same electromagnetic torque. As a consequence, the power factor of the motor decreases as the load increases.

## 3. LTDRO-ADRC Speed Regulation Method

### 3.1. Design of the LTDRO

During sudden load changes, the Extended State Observer (ESO) within the speed loop, which relies on a speed error detection method, may not accurately or promptly estimate the total disturbance. This limitation stems from the fact that the ESO’s bandwidth dictates the quickest rate at which it can track signal changes. A bandwidth set too low hampers the ESO’s ability to swiftly adapt to abrupt load alterations. Conversely, while ESOs commonly employ filters to diminish noise and enhance the smoothness of their estimates, these filters can mitigate fluctuations at the cost of introducing delays, thus decelerating the response to rapid shifts.

To augment the ESO’s responsiveness to sudden load variations, optimizing the design parameters to elevate its bandwidth is a viable approach. Nonetheless, an excessively high bandwidth might provoke issues such as noise amplification and heightened sensitivity, which could destabilize the system. Alternatively, implementing a higher-order ESO design, like a third-order or above, could address these challenges, although it significantly complicates the tuning process.

This paper introduces a novel approach: the design of a load torque dimensionality reduction observer. This observer internally calculates the load torque signal, utilizing it as a compensatory current that is fed forward into the current loop input. It is adept at estimating and compensating for sudden load disturbances—tasks that the conventional ESO might find challenging. Thus, it optimizes the latency issues associated with the ESO in traditional Active Disturbance Rejection Control (ADRC) setups. As a result, it not only accelerates the ADRC speed loop’s response rate but also bolsters the system’s overall adaptability to load changes.

The principle block diagram illustrating the composite ADRC system, which incorporates the load torque dimensionality reduction observer for feedforward compensation, is depicted in Figure 2.

According to the motor’s motion equation, when the motor runs stably under no load, the electromagnetic torque is balanced with the speed, namely:(10)$$T_{e}-B\omega_{m}=0$$

Upon encountering a sudden shift from a zero load condition, the previously established equilibrium between electromagnetic torque and speed becomes disturbed, leading to alterations in the motor’s rotational velocity. At this critical moment, by introducing a compensatory electromagnetic torque, $T_{ec}$, the motor’s motion equation can be reformulated as follows:(11)$$J\frac{d\omega_{m}}{dt}=T_{e}-B\omega_{m}-T_{L}+T_{ec}$$

Assuming the compensatory electromagnetic torque $T_{ec}$ applied to the motor equals the load torque induced by the sudden load change, that is, $T_{ec}=T_{L}$, then the torque and speed remain in equilibrium, and the motor speed remains unchanged. Based on the relationship between electromagnetic torque and current, it is possible to achieve the purpose of compensating the electromagnetic torque of the motor by adjusting the $i_{q}$ current. The compensatory current $i_{qc}$ is as follows:(12)$$i_{qc}=\frac{2}{3\cdot n_{p}\phi_{f}}T_{ec}=\frac{2}{3\cdot n_{p}\phi_{f}}T_{L}$$

Due to the difficulty in directly measuring the load torque, the magnitude of the load torque is indirectly determined based on the motor’s motion equation, utilizing the system’s state variables $i_{q},\;\omega_{m}$. Integrating load torque, speed, and current into one formula, the relationship between them is as follows:(13)$$T_{L}=-\frac{3}{2}n_{p}\phi_{f}i_{q}-B\omega_{m}-J\dot{\omega_{m}}$$

Furthermore, as previously mentioned, the following formula can be derived:(14)$$\left\{\begin{array}{l} T_{e}=J\frac{d\omega_{r}}{dt}+B\omega_{r}+T_{L}\\ \omega_{r}=\frac{d\theta_{m}}{dt} \end{array}\right.$$

Assuming the sampling frequency is sufficiently high, the observed state variable of the load torque can be considered constant within one sampling period, implying that the load torque remains constant during the sampling cycle, i.e., $dT_{l}/dt=0$. By integrating this assumption with the mechanical motion equation, the state equation for a two-dimensional linear time-invariant system is designed as follows:(15)$$\left\{\begin{array}{l} \frac{dx}{dt}=Ax+Bu\\ y=Cx \end{array}\right.$$

Herein, $A=\left[\begin{array}{cc} -\frac{B}{J} & -\frac{1}{J}\\ 0 & 0 \end{array}\right]$, $B=\left[\begin{array}{c} \frac{1}{J}\\ 0 \end{array}\right]$, ${C=\left[\begin{array}{c} 1\\ 0 \end{array}\right]}^{T}$, $x=\left[\begin{array}{c} \omega_{r}\\ T_{L} \end{array}\right]$, ${y=\omega }_{r}$;

It follows that: $\frac{d}{dt}\left[\begin{array}{c} \hat{\omega_{r}}\\ \hat{T_{L}} \end{array}\right]=\left[\begin{array}{cc} -\frac{B}{J} & -\frac{1}{J}\\ 0 & 0 \end{array}\right]\left[\begin{array}{c} \hat{\omega_{r}}\\ \hat{T_{L}} \end{array}\right]+\left[\begin{array}{c} \frac{1}{J}\\ 0 \end{array}\right]T_{e}+\left[\begin{array}{c} k_{1}\\ k_{2} \end{array}\right]\omega_{m}$

Simplifying the above equation and substituting it into the mechanical motion equation yields:(16)$$\left\{\begin{array}{c} \hat{\omega_{r}}=\frac{T_{e}+\omega_{m}\left(k_{1}J-\frac{k_{2}}{s}\right)}{sJ+B}\\ \hat{T_{L}}=\omega_{m}(k_{1}J-\frac{k_{2}}{s}) \end{array}\right.$$

According to the simplified Equation (16), the block diagram of the observer can be obtained, as shown in Figure 3.

In practical applications, to facilitate the design of the observer’s pole placement, the damping coefficient B is considered negligible. By setting $k_{1}J=k_{p}$ and $k_{2}=k_{i}$, the following can be derived:(17)$$\hat{\omega_{r}}=\frac{\left(k_{p}+\frac{k_{i}}{s}\right)\omega_{m}+T_{e}}{sJ}$$

At this juncture, the Dimensionality-Reduced Observer (DRO) can achieve a performance closely paralleling that of the widely recognized Proportional–Integral (PI) controller. In the development and subsequent refinement of the DRO, attention can be exclusively directed towards the proportional and integral components. This approach simplifies the tuning process significantly.

Regarding the decoupled control strategy employed for the current loop, when the motor’s damping coefficient (B) along with the impacts of sampling and filtering present in both the speed and current loops are disregarded, the open-loop transfer function characterizing the stepper motor’s control current loop is delineated as follows:(18)$$G_{c}\left(s\right)=\frac{sK_{Pi}+K_{Ii}}{s\left(sL_{s}+R\right)}$$

In the equation, s is the Laplace transform variable, representing frequency in the complex frequency domain; $K_{Pi}$ is the proportional gain, $K_{Pi}$ can be equivalent to proportional control; $K_{Ii}$ is the integral gain, making up the integral part of the PI controller; $L_{s}$ is the motor inductance, characterizing the electromagnetic response of the motor coils to changes in current; R is the motor resistance, representing the resistance of the motor coils.

To simplify the calculation, assume that $K_{Pi},K_{Ii}$ are related only to the cutoff frequency $f_{c}$ of the current loop, and let ${K_{Pi}=f_{c}L_{s},K}_{Ii}=L_{s}R$. Consequently, the closed-loop transfer function of the current loop can be simplified as follows:(19)$$G_{cl}\left(s\right)=\frac{G_{c}\left(s\right)}{{1+G}_{c}\left(s\right)H\left(s\right)}=\frac{f_{c}}{s+f_{c}}$$

The open-loop transfer function of the velocity loop is:(20)$$G_{\omega }\left(s\right)=\frac{f_{c}{(sk}_{P\omega }+K_{I\omega })}{s\left(s+f_{c}\right)}$$

By simultaneous Equations (19) and (20), the system transfer function can be expressed as:(21)$$G\left(s\right)=\frac{{(sk}_{P}+K_{I})}{s\left(sJ+k_{p}\right)+K_{I}}=G_{o}\left(s\right)/\left(1+G_{o}\left(s\right)\right)$$

The open-loop transfer function $G_{o}\left(s\right)$ can be determined as:(22)$$G_{o}\left(s\right)=\frac{{sk}_{P}+K_{I}}{s^{2}J}$$

As demonstrated by the transfer function depicted in Figure 4, the system exhibits the dynamic properties characteristic of a second-order system. Here, J symbolizes the inertia of the system. Such a configuration is proficient in eliminating steady-state errors while achieving precise tracking of the observed load torque. Nonetheless, the absence of damping contributes to potential oscillations, rendering the system excessively sensitive under specific conditions.

### 3.2. Torque Feedforward Compensation

In motor speed control systems, the conventional application of Proportional–Integral (PI) controllers within the speed loop frequently results in extended adjustment periods following speed variances provoked by alterations in load, thereby complicating the swift minimization of discrepancies between the predetermined speed and its actual counterpart. To augment the system’s dynamic responsiveness to perturbations in load, the introduction of a load torque feedforward compensation strategy is proposed.

This approach entails incorporating a torque current directly into the motor current loop input, which is proportional to the load torque, thereby enabling immediate feedforward compensation for the impact of load disturbances. As depicted in Figure 5, the torque current is determined based on the load torque as estimated by the observer. Following low-pass filtering, it is then injected into the motor current loop, significantly enhancing the motor system’s ability to respond to abrupt changes in load conditions.

### 3.3. Stability Analysis

From the preceding analysis, the system’s transfer function has been derived as $G\left(S\right)$. To analyze stability, it is necessary to define the system’s state space. For simplicity, a reduced model with similar dynamic behavior is considered:(23)$$\left\{\begin{array}{c} \dot{x}=Ax+Bu\\ y=Cx \end{array}\right.$$

Here, x represents the state vector, u the input, and y the output. Incorporating the specific form of the original system’s transfer function, and defining the system state vector as $x={\left[x_{1},x_{2}\right]}^{T}$, the simplified model of the system can be expressed as:(24)$$\left\{\begin{array}{c} {\dot{x}}_{1}=x_{2}\\ \dot{x_{2}}=-\frac{K_{p}}{J}x_{2}-\frac{K_{I}}{J}x_{1}+\frac{1}{J}u \end{array}\right.$$

A Lyapunov function, $V\left(x\right)$, is selected as:(25)$$V\left(x\right)=\frac{x_{1}^{2}}{2}+\frac{x_{2}^{2}}{2}$$

This function is positive definite for x ≠0 and attains its minimum value of 0 when x=0, fulfilling the basic requirements for a Lyapunov function. Further derivation yields the derivative of $V\left(x\right)$ with respect to $\dot{V}\left(x\right)$:(26)$$\begin{array}{l} \dot{V}\left(x\right)=x_{1}\dot{x_{1}}+x_{2}\dot{x_{2}}\\ \;=x_{1}x_{2}+x_{2}\left(-\frac{k_{P}}{J}x_{2}-\frac{K_{I}}{J}x_{1}+\frac{1}{J}u\right)\\ \;=-\frac{k_{P}}{J}x_{2}^{2}-\frac{K_{I}}{J}x_{1}x_{2}+\frac{1}{J}x_{2}u \end{array}$$

Given that $K_{P}$ and $K_{I}$ are non-negative, by designing an appropriate control law, it can be ensured that $\dot{V}\left(x\right)\leq 0$ always holds true, indicating that the system is stable. On the other hand, as derived from the preceding discussion, regarding the system’s poles, it is known that:(27)$$s^{2}J+{sk}_{p}+K_{I}=0$$

This constitutes a quadratic equation in terms of S, with the general solution being:(28)$$s=\frac{-k_{P}+\sqrt{k_{P}^{2}-4JK_{I}}}{2J}$$

The stability of the system necessitates that the real parts of all poles be less than zero. Therefore, it is imperative to examine the real part of the aforementioned solution.

Should the condition $k_{P}^{2}-4JK_{I}>0$ hold, the roots are real numbers. If $-k_{P}<0$, the real part is negative, which implies $k_{P}>0$;

If $k_{P}^{2}-4JK_{I}=0$, there exists a repeated root, upon which the system’s stability also hinges, requiring that $k_{P}>0$;

In the event that $k_{P}^{2}-4JK_{I}<0$, the roots are complex conjugates, with their real parts determined by $\frac{-k_{P}}{2J}$. Under these circumstances, the stability of the system similarly depends on $k_{P}>0$.

In summary, the stability of this system is primarily dependent on the control parameters $k_{P}{,\;K}_{I}$. With appropriate parameter settings, the system is stable.

## 4. Simulation and Experimental Verification

### 4.1. Simulink Simulation

The LTDRO-ADRC control system for a stepper motor was developed and assessed through simulation in Simulink, thereby validating the effectiveness of the control methodology proposed in this investigation.

The parameters defining the chosen stepper motor are delineated in Table 2.

To assess the control performance of the designed controller regarding motor speed, a simulation model of the control system was designed and evaluated within the Matlab/Simulink environment. The simulation employed identical parameters to those utilized in the experimental setup.

To enrich the evaluation of the controller’s efficacy, the speed loop incorporates four distinct controllers for comparative analysis: PI, ADRC, another variant CADRC, and the LTDRO-ADRC proposed in this study. The comparison among these four control systems is depicted in Figure 6.

In the simulation setup, a step speed signal is applied to the stepper motor at 0 s, with a target speed of 50 rpm. The time it takes for the motor to accelerate from zero to the target speed is observed, waiting for the motor to initiate. This procedure facilitates a comparative assessment of the efficacy of various control methodologies. Once the motor’s speed stabilizes, a 1 N·m load is introduced to the test stepper motor at 0.1 s. The variations in speed following the application of this load under different control strategies are monitored. For a more intuitive comparison, all results are consolidated into a waveform graph. After the motor’s speed stabilizes once more, the previously applied 1 N·m load is removed at 0.4 s, and the outcome of this change under each control method is observed. The entire simulation spans 0.5 s. The simulation process and methodology are detailed in Table 3.

The observed torque waveforms in the add and subtract load simulation are shown in Figure 7, which shows that there is an instantaneous torque observed by the LTDRO to exist during the 0–0.05 s time period, even though there is no external load applied, due to the motor starting from zero speed. After the motor speed is stabilized, the load changes at 0.1 s and 0.4 s are accurately observed by the LTDRO. This validates the effectiveness of the LTDRO.

Figure 8 shows the comparison of the response under the use of different control methods. From the figure, it can be seen that there is a contradiction between overshooting and fast response in the PI control method. Here KP=50, KI=10. The PI speed loop can quickly make the motor reach the target speed, but there is overshooting and some oscillation. The peak speed is reached at 0.02127 s with $V_{max}=51.51\;rpm$. Both the ADRC and CADRC have good smoothness and responsiveness. Specifically, the ADRC here has better responsiveness and reaches the target speed in 0.0557 s. This is because the TD in the ADRC has a certain delay for the tracking of the input signal, while the CADRC tends to make a certain trade-off in responsiveness due to the inclusion of other algorithms and observers, as well as the delay of the ESO, and the CADRC reaches the target speed in 0.0656 s. As the LTDRO-ADRC proposed in this paper adopts the principle of the reduced dimension observer, the computational volume is small, and the computing burden of the system is then reduced a lot, sharing part of the task of the ESO. According to the simulation results, the target speed is reached at the moment of 0.0233 s. The performance comparison table of different control methods in the startup phase is shown in Table 4.

After applying the load at 0.1 s, as shown in Figure 9, it can be seen that all the ADRC methods have better anti-jamming ability. The PI control will quickly reduce the speed to 48.74 rpm after applying the load, and then the P controller will pull the speed back to the original speed, but at this time there will be the phenomenon of tiny overshoot, and the speed will be increased to 50.03 rpm. The I controller will pull the speed down to the target speed again, and overshooting and oscillation of the speed will occur under PID control. This is also in line with the flaws of PID control in the previous theoretical analysis. It can be seen from Figure 9a that, after the speed is stabilized again, the different control methods all have certain steady-state errors. The PI enters the steady state at the moment of 0.1158 s, with a steady-state error of −0.27 rpm, while the ADRC and CADRC enter the steady state at the moments of 0.1142 s and 0.1128 s, respectively. The steady-state error is −0.17 rpm and −0.09 rpm, respectively. As shown in Figure 9b, the LTDRO-ADRC proposed in this paper enters the steady state again at the moment of 0.1049 s, and the steady-state error of the speed is −0.01 rpm. It exhibits an excellent anti-interference capability. Table 5 shows the table of performance comparison data of different control methods in the applied load stage.

As shown in Figure 10, the plots comparing the steady-state errors of different control methods are consistent with the data in Table 5.

Figure 11 shows a comparison of the speed change after removing the load for 0.4 s, and it can be seen that the speed of the PID control rises rapidly after removing the load, and there are still oscillations in the regulation process. All ADRC controls have basically no fluctuation in speed after removing the load, but from the specific data, the LTDRO-ADRC has the shortest response time. Detailed data comparisons are shown in Table 6.

In order to better verify the effectiveness of the LTDRO-ADRC, increase and decrease speed control experiments are added. Set the stepper motor startup target speed value as 50 rpm, increase the speed to 70 rpm at 0.2 s, and the speed plummets to zero at 0.3 s. Observe the speed change curve of different control methods. As shown in Figure 12, it can be seen that LTDRO-ADRC has excellent tracking ability compared to other control methods. The speed response is fast and no overshoot occurs.

### 4.2. Experimental Verification

To verify the feasibility of the LTDRO-ADRC proposed in this paper, a stepper motor drive testing platform was built based on a custom closed-loop vector control board, with data collected via serial port and CAN bus for ease of testing and data collection. The control core hardware and its installation are illustrated in Figure 13.

Experiments were carried out using the LTDRO-ADRC speed ring, using the same current loop and control parameters, with the speed command kept at 50 r/min, and a 1 N-m load was applied abruptly at 0.1 s and removed at 0.4 s, respectively. The observer output is shown in Figure 14.

As shown in Figure 15, the current response of the stepper motor is continuous and does not show any abnormal jumps or instability, and the motor is loaded and running within 0.1–0.4 s, and the current reaches a steady state without obvious noise, which indicates that the system can well realize the closed-loop vector control, and verifies the reasonableness and validity of the LTDRO-ADRC for the speed loop.

Ideally, after a sudden load change, the error between the actual speed and the expected speed should be zero. However, under actual working conditions, when the load is increased, the speed will first decrease and then quickly approach the expected speed. The difference between the actual speed and the expected speed after the system stabilizes again is called the steady-state error. The experimental results show that the LTDRO-ADRC can stabilize the speed error in a range closer to the expected value.

In summary, simulation and experiment show that the LTDRO-ADRC speed control method proposed in this study can effectively reduce the fluctuation of stepper motor speed under sudden load change, and improve the system’s anti-interference ability and robustness. It can be well applied in robot ground joints, and can achieve light weight, low cost, and high performance of joint motors.

## 5. Conclusions

To address the traditional stepper motor vector PI speed control system: the internal and external perturbations, such as sudden load changes, and the contradiction between response and overshoot affects the speed control performance, and the ESO in ADRC technology in a high-performance stepper motor system is limited by the arithmetic load, which may lead to delays in the real-time requirements of the motor control; especially in the higher-order system, the complexity of parameter tuning limits the performance of the ADRC problem. Based on the vector control of stepper motors, according to the principle of self-immunity control, a self-immunity control strategy is adopted in the speed loop, and on this basis, a dimensionality reduction observer (DRO) is utilized for load estimation to reduce the system burden of the ESO, and at the same time, the load torque dimensionality reduction observer (LTDRO) internally calculates and outputs the feedforward torque current to be compensated to the input of the current loop. Simulation and experimental results show that the LTDRO-ADRC parameter design is effective and feasible, and the parameter design can be accomplished by using fewer indicators, which reduces the difficulty of design and calculation. This speed control method can effectively improve the dynamic control performance of the system, realize fast response, and improve the system stability at the same time. Simulation results show that the LTDRO-ADRC can stabilize the speed again within 49 ms and 17 ms, respectively, when faced with sudden load increase and sudden load removal. Meanwhile, in terms of steady-state error, compared with the ADRC and CADRC, the improvement is 94% and 88%, respectively. In terms of zero-speed motor starting, the response speed is improved by 58% compared with the traditional ADRC. This provides a solid foundation for applying this control technology to high-performance servo applications.

## Figures and Tables

**Figure 1 sensors-24-02605-f001:**
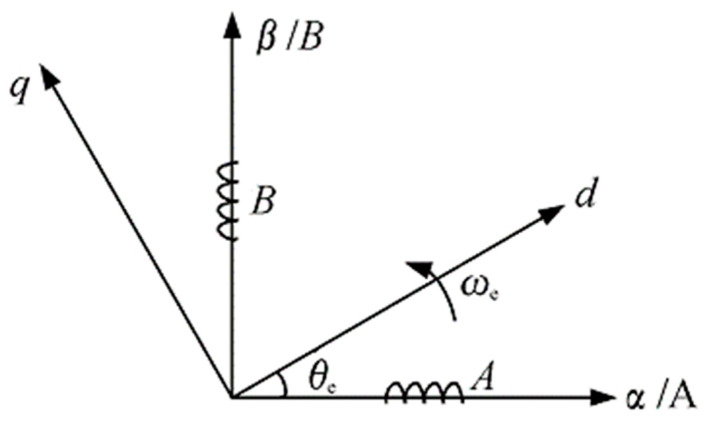
Coordinate transformation relationship diagram.

**Figure 2 sensors-24-02605-f002:**
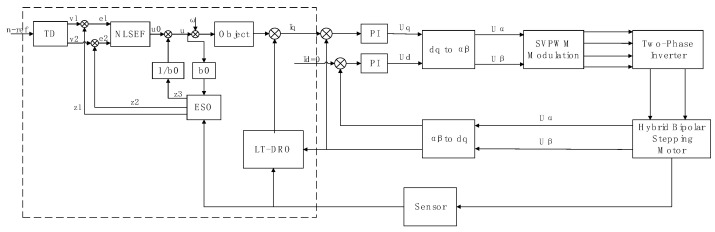
Principle Block Diagram of the LTDRO-ADRC.

**Figure 3 sensors-24-02605-f003:**
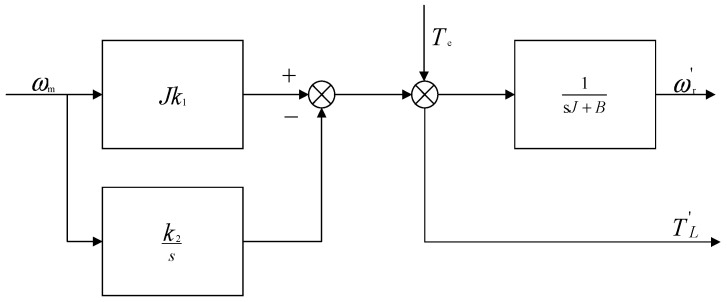
Observer structure diagram.

**Figure 4 sensors-24-02605-f004:**
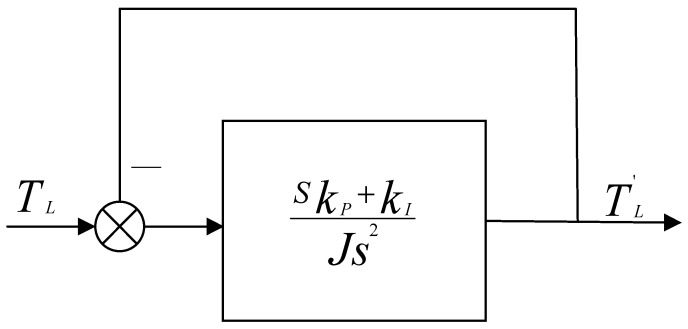
Equivalent System Block Diagram of the LTDRO.

**Figure 5 sensors-24-02605-f005:**
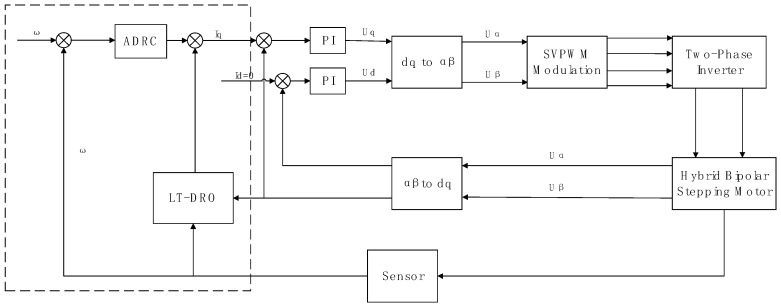
Torque Feedforward Diagram.

**Figure 6 sensors-24-02605-f006:**
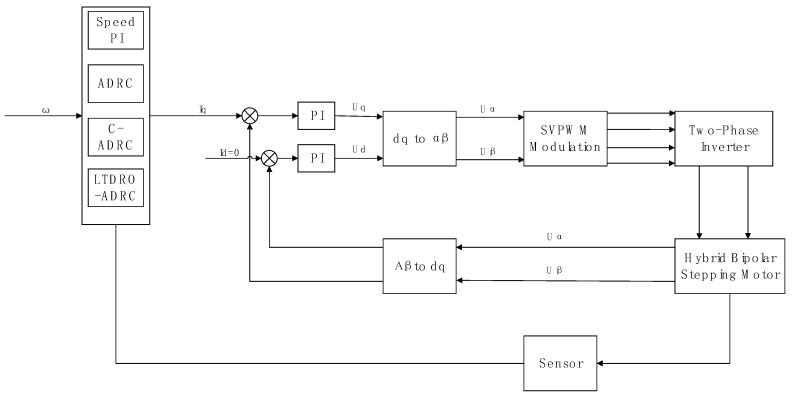
Comparative Block Diagram of Different Speed Loop Controls.

**Figure 7 sensors-24-02605-f007:**
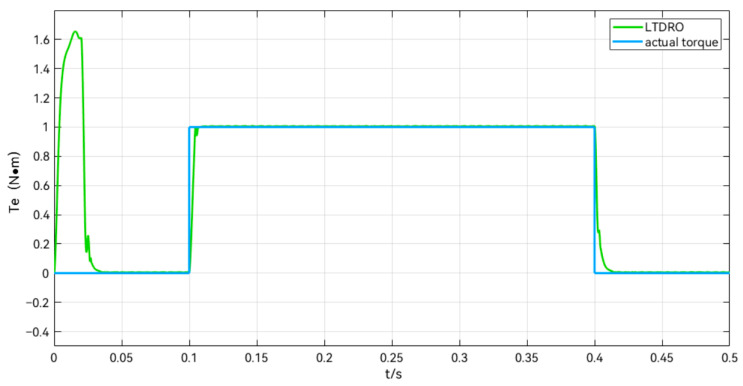
Observed torque diagram during sudden load change.

**Figure 8 sensors-24-02605-f008:**
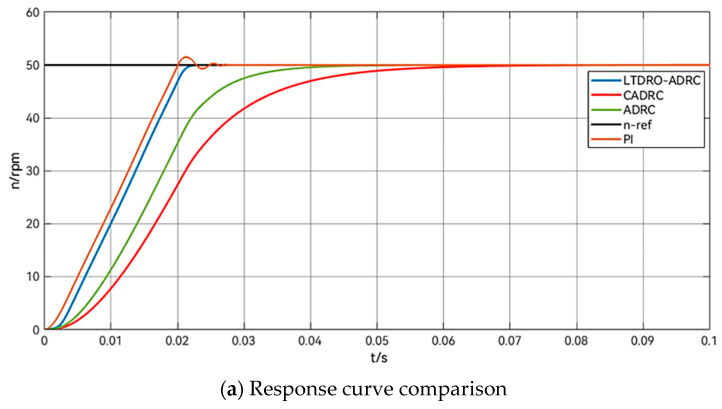
Response diagram of different control methods after starting the motor.

**Figure 9 sensors-24-02605-f009:**
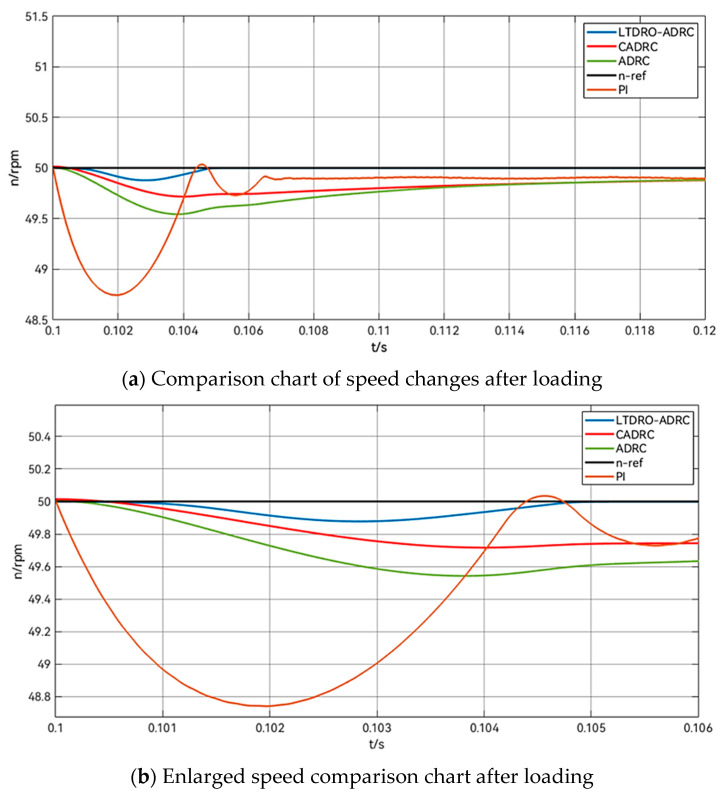
Comparison of speed change after applying load for 0.1 s.

**Figure 10 sensors-24-02605-f010:**
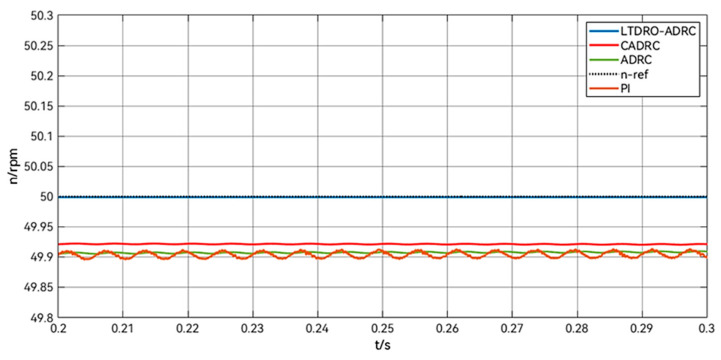
Comparison of velocity steady-state error in 0.2–0.3 s.

**Figure 11 sensors-24-02605-f011:**
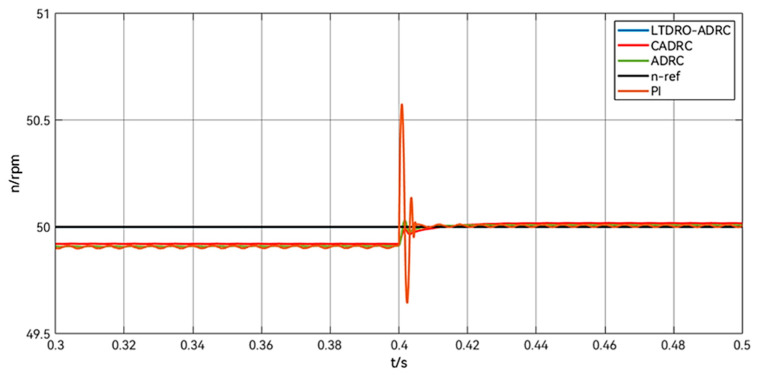
Comparison of speed change after unloading at 0.4 s.

**Figure 12 sensors-24-02605-f012:**
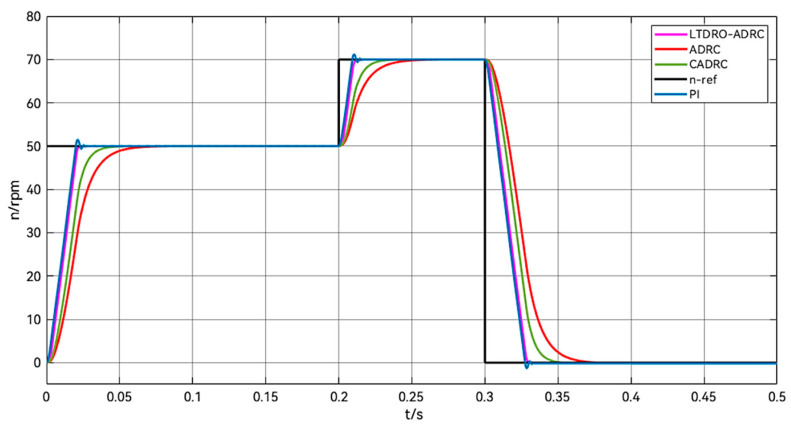
Comparison of changes in acceleration and deceleration control.

**Figure 13 sensors-24-02605-f013:**
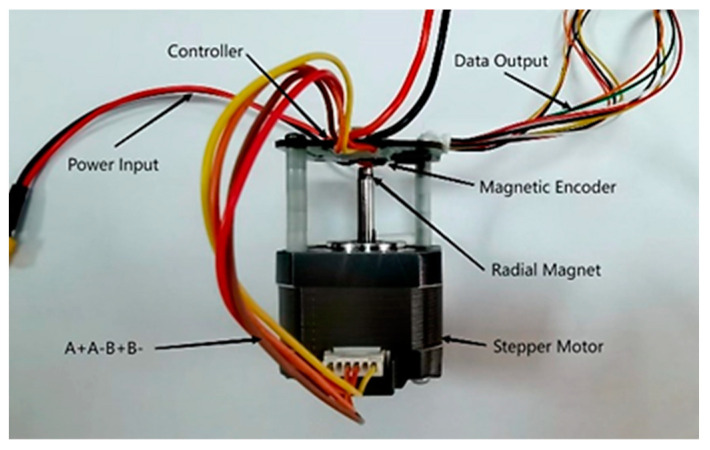
Core Hardware of the Testing Platform.

**Figure 14 sensors-24-02605-f014:**
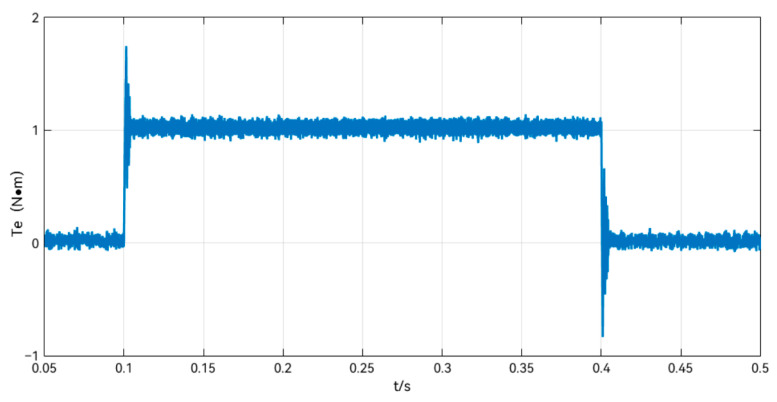
LTDRO Output Waveform.

**Figure 15 sensors-24-02605-f015:**
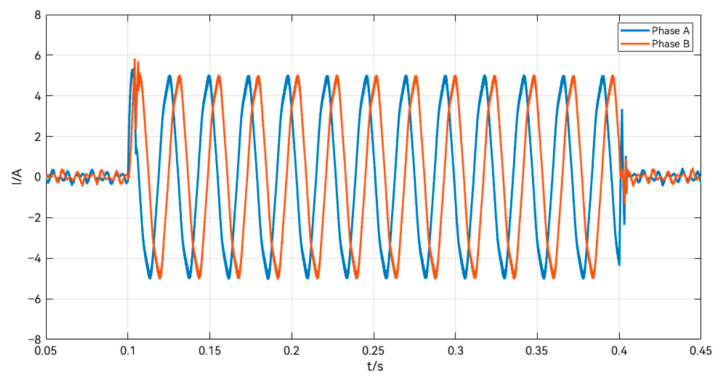
Output current of the LTDRO-ADRC.

**Table 1 sensors-24-02605-t001:** Summary of Current Research on ADRCs.

Method Category	Techniques/Algorithms
Parameter Optimization	GA, PSO/IPM, APSO
Composite Design	MPC, Luenberger, SM-ADRC, SADRC, Fuzzy-ADRC
Structural Optimization	IESO, NPLLO, RESO
Algorithm Combination	TD3, IMA, Interpolation Fitting

**Table 2 sensors-24-02605-t002:** Main parameters of stepper motor.

Parameter	Value
Winding inductance	0.0042 H
Winding resistance	2.10 hm
Step angle	1.8°
Maximum flux linkage	0.00424 Vs
Maximum detent torque	0.065 N·m
Total inertia	$0.0058\;kg\cdot m^{2}$
Total friction	0.0013 N·m·s
Number of pole pairs	50

**Table 3 sensors-24-02605-t003:** Load Application and Removal Simulation Procedure.

Time Node	Operation
0 s	Start motor, V=50 rpm
0.1 s	Apply load, $T_{e}=1\;N\cdot m$
0.4 s	Remove load, $T_{e}=0\;N\cdot m$
0.5 s	Simulation over

**Table 4 sensors-24-02605-t004:** Comparative performance data for the startup phase.

Time Node	Control Method	Peak/Valley	Response Moment	Error
Start-up phase	PI	51.51/49.34	0.0275	1.51
Start-up phase	ADRC	50	0.0557	0
Start-up phase	CADRC	50	0.0656	0
Start-up phase	LTDRO-ADRC	50	0.0233	0

**Table 5 sensors-24-02605-t005:** Performance Comparison Data for Applied Load Stage.

Time Node	Control Method	Peak/ Valley	Response Moment	Steady-State Error	Steady-State Moment
Load application	PI	48.74	0.1019	−0.27	−0.27
		50.03	0.1046		
Load application	ADRC	49.54	0.1133	−0.17	0.1142
Load application	CADRC	49.72	0.1038	−0.09	0.1128
Load application	LTDRO-ADRC	49.98	0.1029	−0.01	0.1049

**Table 6 sensors-24-02605-t006:** Comparative performance data for the unloading phase.

Time Node	Control Method	Peak/ Valley	Response Moment	Steady-State Error	Steady-State Moment
Remove load	PI	50.57	0.4009	0.01	0.4147
		49.64	0.4024		
Remove load	ADRC	50.03	0.4017	0	0.4091
Remove load	CADRC	49.99	0.4098	0	0.4119
Remove load	LTDRO-ADRC	50	0.4009	0	0.4017

## Data Availability

Data are contained within the article.

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
