# Peer review of "Composite ADRC Speed Control Method Based on LTDRO Feedforward Compensation"

_sensors, 2024, doi:10.3390/s24082605_

Round 1

Reviewer 1 Report

Comments and Suggestions for Authors

In the present manuscript, an Active Disturbance Rejection Control (ADRC) with the Load Torque Dimensionality Reduction Observer (LTDRO) has been presented for stepper motor control. The simulation and experimental results show that the proposed LTDRO-ADRC parameter design is effective and feasible. The subject of the paper is of scientific interest and the work presented contributes to the knowledge in the area of active vibration control.

However, the submission could not be recommended for acceptance in Sensors because this work is based on well-known principles with little contribution in terms of either new methods or new physical insights. As such, its contribution to the existing knowledge is marginal.

    The authors give the control performance by using ADRC and PI controller. However, the control performance based on ADRC is similar to PI control (as shown in Figs. 11 and 13), according to my experience, the control performance based on ADRC is much better than PI control, which can also be found in lots of published papers. I think the authors may not give the correct ADRC parameters in simulation. In my opinion, if ADRC with correect parameters, the control performance could be similar to authors‘ proposed method.

Author Response

The revised manuscript, along with our responses to your insightful comments, can be found in the attached document. Changes made to the original text are highlighted in red for ease of identification.

Reviewer 2 Report

Comments and Suggestions for Authors

This paper proposes a composite ADRC method for stepper motors combined 114 with the Load Torque Dimensionality Reduction Observer (LTDRO). 

The results are presented clearly, the analysis is well built-up. The conclusions are correct and adequate.

A few notes:

- The expression on the left side of formula 11 is incorrect: B does not have to appear here, and the time derivative of the angular velocity must be written.

- Formula 17 is also incorrect: "Te" should not be multiplied by the expression in parentheses.

Author Response

(The authors gave the same response as above.)

Reviewer 3 Report

Comments and Suggestions for Authors

This paper proposes a composite ADRC method for stepper motors combined with the Load Torque Dimensionality Reduction Observer (LTDRO). The observer estimates external load disturbances in real time and outputs a feedforward compensation current after internal calculation. ESO is used to compensate for all disturbances except load torque.  Simulation and experimental results are presented to demonstrate the effectiveness. 

The comments are listed as follows.

1. The obtained results in Abstract should be more detailed numerically, rather than “demonstrate the effectiveness.

2. The paper employs a LTDRO to to estimate the abrupt load disturbances to overcome the high real-time requirement in ESO. However, it is unclear how it works in principle. The motivation should be stated in Introduction.

3. Regarding reduced-order ESO (RESO) in Introduction, the works “Structural Vibration Suppression Using a Reduced-Order Extended State Observer-Based Nonsingular Terminal Sliding Mode Controller with an Inertial Actuator”, “Model-Assisted Reduced-Order ESO Based Command Filtered Tracking Control of Flexible-Joint Manipulators with Matched and Mismatched Disturbances” are suggested to cited and make a analysis.

4. Line 142, it is unclear what are the two coordinates?

5. All the figures presented in the paper are too blurry.

6. The simulation results show that the steady-state error is reduced to a mere 0.03%. What is the physical meaning of a mere 0.03%?

7. How about the stability issue of the whole control system?

8. The conclusions should be refined.

Comments on the Quality of English Language

Author Response

(The authors gave the same response as above.)

Round 2

Reviewer 1 Report

Comments and Suggestions for Authors

After examining the revised manuscript, I find the authors have faithfully responded according to my comments. The revised article is recommended for publication in its present form.

Reviewer 3 Report

Comments and Suggestions for Authors

My questions have been carefully addressed. No more questions, and it can be accepted.